# Analgesic Effect of Boiogito, a Japanese Traditional Kampo Medicine, on Post-Traumatic Knee Osteoarthritis through Inhibition of ERK1/2 Phosphorylation in the Dorsal Horn of the Spinal Cord

Yusuke Kunieda [1], Takayuki Okumo [1,2,*], Hideshi Ikemoto [1], Naoki Adachi [1], Midori Tanaka [1], Taro Kimura [1,2], Kanako Yusa [1,2], Koji Kanzaki [2] and Masataka Sunagawa [1]

[1]  Department of Physiology, School of Medicine, Showa University, 1-5-8 Hatanodai, Shinagawa-ku, Tokyo 142-8555, Japan; situation.right.here@gmail.com (Y.K.); h_ikemoto@med.showa-u.ac.jp (H.I.); nadachi@med.showa-u.ac.jp (N.A.); midori.tanaka625@gmail.com (M.T.); volttarou@med.showa-u.ac.jp (T.K.); kanakotama21@gmail.com (K.Y.); suna@med.showa-u.ac.jp (M.S.)
[2]  Department of Orthopedic Surgery, Showa University Fujigaoka Hospital, 1-30 Fujigaoka, Aoba-ku, Yokohama City 227-8501, Japan; kkanzaki@med.showa-u.ac.jp
*  Correspondence: tokumo@med.showa-u.ac.jp; Tel.: +81-3-3784-8110

**Abstract:** Boiogito (BO), a Japanese traditional herbal medicine, has been proven to be clinically effective against knee osteoarthritis (KOA)-associated pain. However, the therapeutic mechanism of BO remains unclear. Thus, we investigated the analgesic mechanism of BO using a rat KOA model. KOA was induced by destabilization of the medial meniscus (DMM). Rats were allocated into the following four groups: control, sham, DMM, and DMM + BO groups. Rotarod test was performed to evaluate the pain-related locomotive dysfunction. Expression of phosphorylated extracellular signal-regulated kinase1/2 (pERK1/2) in the spinal dorsal horn was examined using immunofluorescence staining and Western blotting on days 1 and 28 after DMM surgery. A mitogen-activated protein kinase inhibitor, U0126, was intrathecally injected and rotarod test and Western blotting were performed. The rotarod test revealed hampered locomotive function in the DMM group, which was significantly improved upon BO administration. The number of pERK1/2-positive cells was increased in the DMM group, whereas it was significantly decreased in the DMM + BO group. U0126 significantly inhibited ERK1/2 phosphorylation and increased walking time in the rotarod test, suggesting that the DMM-related pain was associated with ERK1/2 phosphorylation in the spinal dorsal horn. In conclusion, BO administration improved the pain-related locomotive dysfunction by suppressing ERK1/2 phosphorylation.

**Keywords:** osteoarthritis; destabilization of the medial meniscus (DMM); kampo; boiogito; pain; ERK

## 1. Introduction

Knee osteoarthritis (KOA) is a chronic joint disease that affects more than 300 million people worldwide and results in a limited range of motion and dysfunction of the knee joint with alteration in the alignment of the knee joint [1]. Obesity, aging, metabolic syndrome, inheritance, trauma, and infection are the risk factors for KOA [2–4]. The pathophysiology of KOA involves not only the "wear and tear principle" in the articular cartilage [5] but also the disorders of the peri- or intra-articular tissue caused by synovitis, sclerosis of the subchondral bone, and contracture of the ligamentous structure [6]. KOA is known to impair activities of daily living due to pain and swelling associated with chronic inflammation and degenerative changes in the articular joints [7,8].

Surgical and nonsurgical treatments are available for KOA. Surgical treatments such as around knee osteotomy and total knee arthroplasty are performed for terminal KOA [1]. Although they are beneficial with excellent pain-relieving effects and functional recovery,

they are not absolutely recommended for everyone due to their large physical and economic burden. In some cases, postoperative pain becomes chronic, and not all patients are completely satisfied [9]. The Osteoarthritis Research Society International (OARSI) strongly recommends arthritis education, land-based exercise, and mind–body exercise such as tai-chi and yoga as a conservative treatment for KOA [10]. However, the current KOA conservative treatment only alleviates clinical symptoms and cannot improve the pathophysiological condition. Although various disease-modifying osteoarthritis drugs have undergone clinical trials [6], none have been clinically used. In fact, there is a need for treatments that have both an analgesic effect and a preventive effect on KOA.

Studies have reported that herbal medicines reduce inflammation in the knee joint and suppress the progression of KOA [11,12]. Boiogito (BO), a traditional Japanese herbal medicine (Kampo medicine), has been found to be effective for patients with swelling and pain of the knee, and its prescription has been approved by the Japanese Ministry of Health, Labor and Welfare [13]. According to oriental medicine, BO improves water metabolism and eliminates swelling and pain in the knee joint [14] and hence may prevent KOA. Oral administration of BO has been found to relieve gait pain in patients with KOA, and some basic research has reported that it suppresses the inflammatory response in the knee joint [15,16]. We had earlier clarified that BO suppressed the progression of KOA in rats with surgically induced KOA; moreover, BO could also improve locomotive function in the early stage of the disease [17]. The maintenance of motor function despite the induction of KOA suggests that BO has an analgesic effect. However, the mechanism underlying this analgesic effect remains unclear.

The pain associated with KOA has been classified as chronic in which dull pain or a throbbing sensation is exacerbated by long-term morbidity [18]. The development of chronic pain is involved in pain sensitization in peripheral tissues and/or the central nervous system, and it has been reported that pain-related stimuli are also enhanced in the spinal cord in patients with KOA [19]. One pathology underlying central sensitizations is the phosphorylation of extracellular signal-regulated kinase (ERK)1/2 in the dorsal horn of the spinal cord. Pain stimuli in the peripheral tissues ascend myelinated Aδ fibers and unmyelinated C fibers and are then projected onto layers I and II in the dorsal horn of the spinal cord [20]. Neurotransmission in the spinal cord is enhanced by the activation of ERK1/2 [21]. The number of phosphorylated ERK1/2 (pERK1/2)-positive cells was found to increase in rats with hyperalgesia and allodynia [20,21]. In this present study, we examined the analgesic effect of BO on acute postoperative pain and osteoarthritis-associated pain as well as the involvement of ERK1/2 underlying the effect. We made two hypotheses and investigated them. First, BO suppresses the ERK phosphorylation in the dorsal horn of the spinal cord, and second, inhibition of the ERK phosphorylation in the spinal dorsal horn plays an important role in analgesia against KOA-associated pain.

## 2. Materials and Methods

### 2.1. Animals

Twelve-week-old male Wistar rats (Nippon Bio-Supp. Center, Tokyo, Japan) were used in this study. Two to three rats per cage were housed in an animal room under controlled conditions (12 h light/dark cycle, temperature 20–25 °C, and humidity 50–60%) and fed with standard powdered rodent chow (CE-2; CLEA Japan, Tokyo, Japan) and water ad libitum. In this study, a total of 50 rats, including 40 rats in Experiment 1 and 10 rats in Experiment 2, were used. All experimental procedures were approved by the Institutional Ethics Committee for Care and Use of Animals of Showa University (certificate number: 03053, date of approval: 1 April 2021).

### 2.2. Kampo Medicine, BO

The dry powdered extract of BO was supplied by Tsumura & Co. (TJ-20; Lot No. 2190020010, Tokyo, Japan). BO has been approved as a prescription drug by the Japanese Ministry of Health, Labor, and Welfare. BO contains a dry extract of a mixed drug substance

consisting of 5.0 g of *Sinomenium* stem, 5.0 g of *Astragalus* root, 3.0 g of *Attractylodes lancea* rhizome, 3.0 g of Jujube, 1.5 g of *Glycyrrhiza*, and 1.0 g of ginger. These herbs are extracted by mixing with purified water at 95.1 °C for 1 h, and the soluble extract is separated from the insoluble residue, and dried by removing water under reduced pressure. All crude drugs are listed in the 17th edition of the Japanese Pharmacopoeia [22]. These varieties, sites, and the proportion of the main pharmacological active ingredients contained in each crude drug are strictly define.

### 2.3. KOA Induction by Destabilization of the Medial Meniscus (DMM)

Meniscal dysfunction due to meniscal tear has been identified as one of the primary causes of post-traumatic KOA, and a similar pathophysiology can also be observed using the surgical procedure of DMM in a rodent animal model [23,24]. We performed DMM and sham surgeries on the right knee under isoflurane (Fujifilm Wako Pure Chemical Corp., Osaka, Japan) inhalation general anesthesia as described previously [17]. Briefly, a 2-cm longitudinal skin incision and a medial parapatellar intervention were performed to observe the articular joint space and medial meniscus. The medial meniscotibial ligament (MMTL) was observed behind the patellar tendon. The MMTL and the medial meniscotibial joint capsule were horizontally transected to induce meniscal destabilization. In the sham group, only the medial joint opening was performed. Finally, the subcutaneous soft tissue was repaired by suturing with 6-0 Vicryl®(Ethicon Inc., Somerville, NJ, USA).

### 2.4. Rotarod Test

The rotarod test was used to verify rodent motor coordination and fatigue tolerance in neurodegenerative disorders such as Parkinson's disease [25] and to validate locomotive function and pain sensitization [26]. In this study, we used an automated accelerating rotarod apparatus with a lane width of 75 mm and a rod diameter of 60 mm (LE8305, Panlab Harvard Apparatus, Barcelona, Spain). Rats were trained for 5 min per day for 2 days to adapt to the experiment. The rotating drum was accelerated at 5–40 rpm over 30 s. The latency to fall off the rotarod apparatus was then recorded. Each rat was subjected to three consecutive trials, and data were recorded as an average of the three trials. The cutoff time was set at 45 s.

### 2.5. Experiment 1: Investigation of the Analgesic Effect of BO

We conducted two independent experiments. In Experiment 1, the analgesic effect of BO was examined. In total, 20 rats aged 12 weeks were divided into the following four groups: control, sham, DMM, and DMM + BO. The dry powdered extract of BO was mixed with powdered rodent chow at a concentration of 3% and fed to rats in the DMM + BO group. The concentration of BO was determined according to previous reports [17,27]. During 28 days after DMM or sham surgery, the rotarod test was performed before and 1, 3, 7, 14, 21, and 28 days after DMM surgery. The spinal cord on L2-L4 was harvested for immunofluorescence staining and western blotting on days 1 and 28 (Figure 1).

### 2.6. Experiment 2: Contribution of ERK1/2 Phosphorylation to Pain-Related Locomotive Dysfunction with DMM Surgery

Intrathecal catheter injection of the mitogen-activated protein kinase (MEK) inhibitor U0126 (Fujifilm Wako Pure Chemical Corp.) was performed to block ERK1/2 phosphorylation. U0126 was dissolved in 20% dimethyl sulfoxide (DMSO, 043-07216; Wako Pure Chemical Industries Ltd., Osaka, Japan) [28]. After the rotarod test, the lumbar area was shaved and sterilized by applying povidone-iodine (Meiji Seika Pharma Co., Tokyo, Japan) under isoflurane inhalation general anesthesia. A small longitudinal skin incision was made on the L5-L6 vertebrae, and a catheter tube (PE-10 tube; Eicom Co., Kyoto, Japan) was inserted into the intrathecal space to the lumbar enlargement of the spinal cord. According to a previous report [21], U0126 (5 µg/10 µL) was intrathecally administered 20 min before DMM surgery (DMM + U0126 group), and 10 µL of 20% DMSO was injected as vehicle control (DMM + vehicle group). After the injection, the PE-10 catheter was removed, and

the supraspinous ligament and subcutaneous soft tissue were repaired by suturing with 6-0 Vicryl®. At 24 h after surgery, the rotarod test was performed, and the spinal cord on L2-L4 was harvested for Western blotting to measure ERK1/2 phosphorylation (Figure 2).

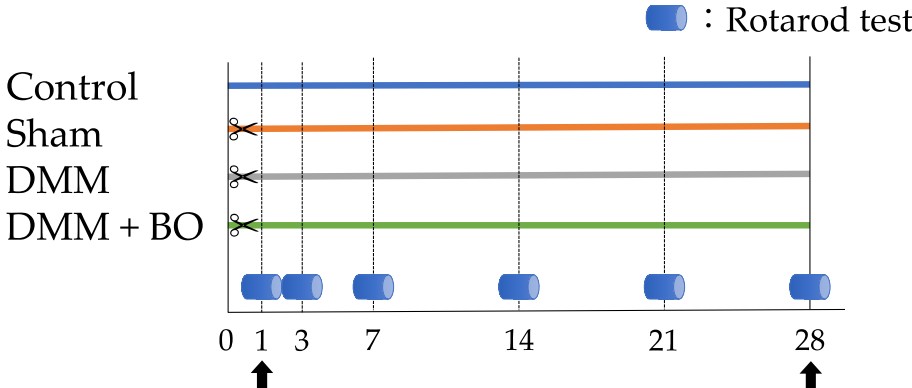

**Figure 1.** Protocol of Experiment 1. Twenty 12-week-old rats were divided into the following four groups: control, sham, DMM, and DMM + BO. During 28 days after DMM or sham surgery, the rotarod test was performed before and 1, 3, 7, 14, 21, and 28 days after DMM surgery. The spinal cord at the L2–L4 level was harvested for immunofluorescence staining and Western blotting on days 1 and 28 after the surgery. DMM, destabilization of the medial meniscus; BO, boiogito.

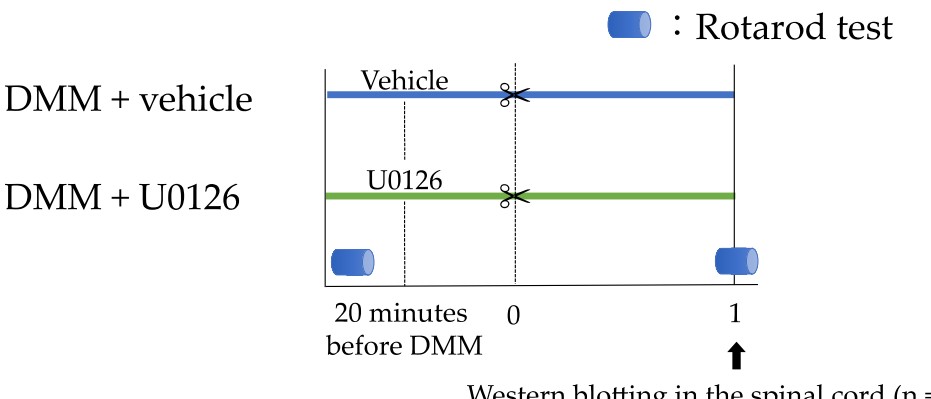

**Figure 2.** Protocol of Experiment 2. Contribution of ERK1/2 phosphorylation to pain-related locomotive dysfunction with DMM surgery was examined. After the rotarod test, U0126 (5 μg/10 μL) was intrathecally administered 20 min before the DMM surgery (DMM + U0126 group), and 10 μL of 20% DMSO was injected as a vehicle control (DMM + vehicle group). One day after the surgery, the spinal cord at the L2–L4 level was harvested for Western blotting to measure ERK1/2 phosphorylation. ERK, extracellular signal-regulated kinase; DMM, destabilization of the medial meniscus, DMSO, dimethyl sulfoxide.

*2.7. Immunofluorescence Staining*

After deep anesthesia by intraperitoneal injection of sodium pentobarbital (Somno-Pentyl, Kyoritsu Seiyaku, Tokyo, Japan), rats were perfused with phosphate-buffered saline (PBS, pH 7.4) and then with 4% paraformaldehyde dissolved in 0.1 M PBS. The excised spinal cord was stored in 4% paraformaldehyde overnight and then immersed in 20% sucrose solution for 48 h. Tissue was embedded in Tissue-Tek optimum cutting temperature (OCT) compound (Tissue-Tek Sakura Finetek, Tokyo, Japan) and stored at −80 °C until use. Each sample was cut into 20-μm-thick sections at the L3 level of the spinal cord using a cryostat (CM1860; Leica Biosystems, Nussloch, Germany). Sections were washed three times with PBS and incubated in PBS containing 10% goat serum and 0.5% Triton-X (Sigma-Aldrich Japan K.K., Tokyo, Japan) for 2 h for blocking and

permeabilization. The sections were then incubated overnight with rabbit pERK anti-body (1:1000, #4370S, Cell Signaling Technology, Beverly, MA, USA) at 4 °C following with the fluorescence-tagged sec-ondary antibody anti-rabbit Alexa Fluor 555 (1:1000, #A31572 Thermo Fisher Scientific, Waltham, MA, USA) for 2 h. Sections were rinsed three times with PBS, and nuclei were stained with 4′,6-diamidino-2-phenylindole (DAPI, 1:1000, Thermo Fisher Scientific) for 10 min. The number of pERK-positive cells in the right dorsal horn of the spinal cord was counted by obtaining fluorescent images of the sections using a confocal laser scanning fluorescence microscope (FV1000D, Olympus, Tokyo, Japan).

*2.8. Western Blotting*

After deep anesthesia with isoflurane inhalation, the rats were euthanized, and a right dorsal part of the L2–L4 spinal cord was immediately harvested and frozen in liquid nitrogen. The tissues were homogenized in lysis buffer containing 1% sodium dodecyl sulfate (SDS), 20 mM Tris-HCl (pH 7.4), 5 mM ethylenediaminetetraacetic acid (EDTA) (pH 8.0), 10 mM sodium fluoride, 2 mM sodium orthovanadate, 0.5 mM phenyl arsine oxide, 1 mM homogenized in lysis buffer containing phenylmethylsulphonyl fluoride. The homogenate was centrifuged at 15,000 rpm for 20 min at room temperature, and the supernatant was collected; the concentration of all samples was standardized using the BCA protein assay kit (Thermo Fisher Scientific). Samples containing an equivalent amount of proteins (12 µg) were subjected to sodium dodecyl sulfate-polyacrylamide gel electrophoresis (SDS-PAGE, 10% SDS) and then transferred onto polyvi-nylidene difluoride (PVDF) membranes. These membranes were then immersed in 5% ($w/v$) bovine serum albumin (#011-21271, FUJIFILM Wako Pure Chemical) containing Tris-buffered saline buffer and Tween 20 (TBST; Sigma-Aldrich Japan Co.) for 1 h at room temperature. After washing the membranes three times with TBST, they were incubated over-night at 4 °C with an anti-pERK1/2 antibody (1:1000, #4370S, Cell Signaling Technology) and anti-ERK1/2 antibody (1:2000, #9102, Cell Signaling Technology). The membranes were then washed with TBST and incubated with goat an-ti-rabbit secondary antibody conjugated with horseradish peroxidase (1:1000, #611-1302, Rockland Immunochemicals, Gilbertsville, PA, USA) for 1 h at room temperature. Chemiluminescence images using a Pierce™ ECL western blotting sub-strate (Thermo Fisher Scientific) were captured with a charge-coupled device camera system (Ez-Capture MG, Atto Co., Tokyo, Japan). The immunoreactivity of each band was quantified using the Lane & Spot Analyzer software (Atto Co., Ltd.). The fold change in the density of pERK in each sample was normalized to that of total ERK.

*2.9. Statistical Analysis*

All experimental data were presented as mean ± standard deviation (SD) of multiple repeats of the same procedures. Statistical analysis was conducted using the JMP® Pro version 14.0 software (SAS Inc., Cary, NC, USA). In Experiment 1, each group was compared via one-way analysis of variance following the verification of the test of normality and the Tukey–Kramer method; in Experiment 2, Welch's *t*-test was performed; *p*-values of < 0.05 were considered statistically significant.

### 3. Results

*3.1. Experiment 1: Analgesic Effect of BO*

Improvement of locomotive function through BO administration was evaluated using the rotarod test. In the acute phase (on days 1 and 3), the latencies to fall off the rotarod apparatus (walking time) became significantly lower in the DMM group than in the control group ($p < 0.01$). However, this decrease was inhibited in the DMM + BO group ($p < 0.05$). In the chronic phase (on day 14 and later), the walking time on the rotarod apparatus in the DMM group was significantly lower than that in the control and sham groups ($p < 0.01$) These decreases were significantly inhibited in the DMM + BO group (on days 14 and 21; $p < 0.01$, on day 28; $p < 0.05$). Moreover, the walking time on day 28 in the DMM + BO group showed no significant difference compared with that in the control group (Figure 3).

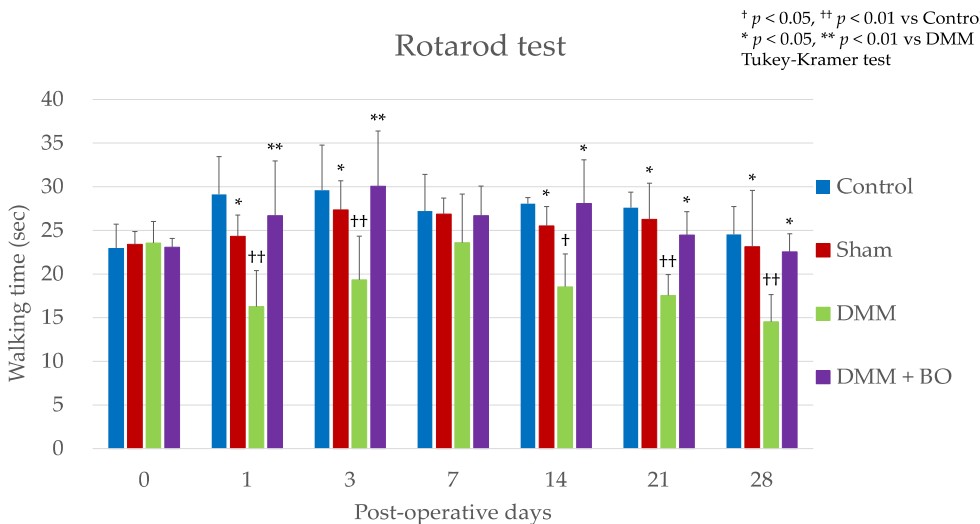

**Figure 3.** Rotarod test. All data are presented as mean $\pm$ standard deviation ($n$ = 5 per group). † $p < 0.05$, †† $p < 0.01$ vs. Control, * $p < 0.05$, ** $p < 0.01$ vs. DMM. DMM, destabilization of the medial meniscus; BO, boiogito.

*3.2. Suppressive Effect of BO on ERK1/2 Phosphorylation in the Dorsal Horn of the Spinal Cord in the Acute Phase after DMM Surgery*

The spinal cord was harvested 1 day after DMM surgery in order to analyze the degree of ERK1/2 phosphorylation. In the DMM group, immunofluorescence staining showed a significant increase in the number of pERK1/2-positive cells in the dorsal horn of the spinal cord compared to that in the control group (7.7 $\pm$ 2.5 vs. 3.3 $\pm$ 0.8, $p < 0.05$). After the oral administration of BO, a significantly lower number of pERK1/2-positive cells was noted in the dorsal horn of the spinal cord (7.7 $\pm$ 2.5 vs. 3.3 $\pm$ 0.8, $p < 0.05$) (Figure 4A,B). Furthermore, the results of Western blotting revealed an increased pERK1/2 expression, which was significantly blocked by BO administration ($p < 0.05$) (Figure 4C,D).

*3.3. Suppressive Effect of BO on ERK1/2 Phosphorylation in the Dorsal Horn of the Spinal Cord in the Chronic Phase after DMM Surgery*

The spinal cord was also harvested 28 days after the DMM surgery. In the DMM group, immunofluorescence staining revealed a significant increase in the number of pERK1/2-positive cells in the dorsal horn of the spinal cord compared to that in the control group (9.3 $\pm$ 2.7 vs. 2.5 $\pm$ 1.1, $p < 0.05$). Oral administration of BO resulted in a significantly lower number of pERK1/2-positive cells in the dorsal horn of the spinal cord than that in the DMM group (9.3 $\pm$ 2.7 vs. 5.6 $\pm$ 1.6, $p < 0.05$) (Figure 5A,B). The results of Western blotting also showed significantly increased pERK1/2 expression, which was blocked by BO administration ($p < 0.05$) (Figure 5C,D).

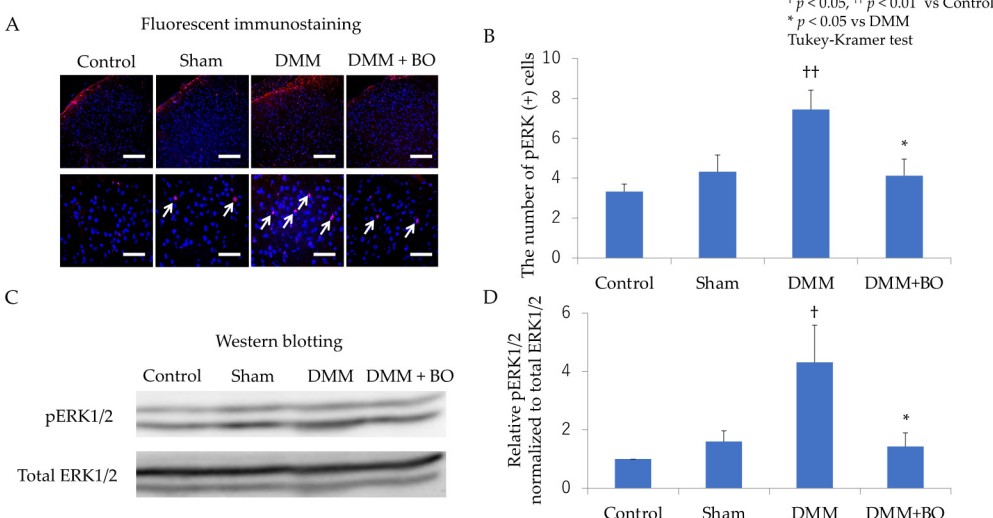

**Figure 4.** Expression of pERK1/2 in the right dorsal horn of the spinal cord in the acute phase (1 day after DMM surgery). (**A**) Representative images of immunofluorescence staining in each group (red; pERK1/2, blue; nucleus). Upper images; magnification: ×200, Scale bar = 150 μm. Lower images; magnification: ×600, Scale bar = 50 μm. White arrow (↗□) indicates a pERK1/2-positive cell. (**B**) The numbers of pERK1/2-positive cells. Each *n* = 5, †† *p* < 0.01 vs. Control, * *p* < 0.05 vs. DMM. (**C**) Immunoblot images of pERK1/2 and total ERK1/2 in the right dorsal horn of the spinal cord. (**D**) The quantified data of the immunoblot for pERK1/2 normalized to total ERK1/2. Each *n* = 5, † *p* < 0.05 vs. control, * *p* < 0.05 vs. DMM. All data are presented as mean ± standard deviation. pERK, phosphorylated extracellular signal-regulated kinase; DMM, destabilization of the medial meniscus; BO, boiogito.

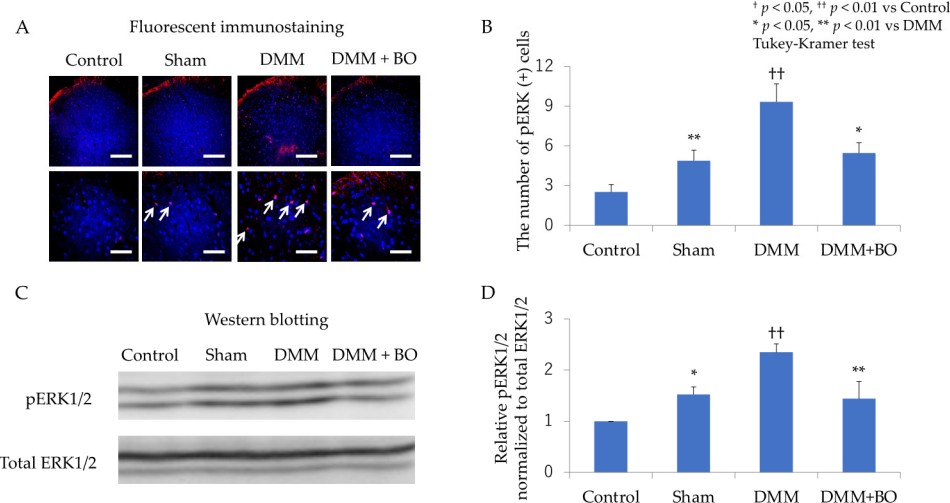

**Figure 5.** Expression of pERK1/2 in the right dorsal horn of the spinal cord in the chronic phase (28 days after DMM surgery). (**A**) Representative images of immunofluorescence staining in each group (red, pERK1/2; blue, nucleus). Upper images; magnification: ×200, Scale bar = 150 μm. Lower images; magnification: ×600, Scale bar = 50 μm. White arrow (↗□) indicates a pERK1/2-positive cell. (**B**) The numbers of pERK1/2-positive cells. Each *n* = 5, †† *p* < 0.01 vs. control, * *p* < 0.05, ** *p* < 0.01 vs. DMM. (**C**) Immunoblot images of pERK1/2 and total ERK1/2 in the right dorsal horn of the spinal cord. (**D**) The quantified data of the immunoblot for pERK1/2 normalized to total ERK1/2. Each *n* = 5, †† *p* < 0.01 vs. Control, * *p* < 0.05, ** *p* < 0.01 vs. DMM. All data are presented as mean ± standard deviation. pERK, phosphorylated extracellular signal-regulated kinase; DMM, destabilization of the medial meniscus; BO, boiogito.

### 3.4. Experiment 2: Contribution of ERK1/2 Phosphorylation to Pain-Related Locomotive Dysfunction with DMM Surgery

The results of Western blotting revealed significant blockade of ERK1/2 phosphorylation in the DMM + U0126 group ($p < 0.05$) (Figure 6A,B). In the rotarod test, the latencies to fall off the rotarod apparatus were not different between the DMM + vehicle group and the DMM + U0126 group before the surgical intervention. After 24 h, rats in the DMM + U0126 group were observed to exhibit significantly longer walking time on the rotarod apparatus than rats in the DMM + vehicle group ($p < 0.05$) (Figure 6C).

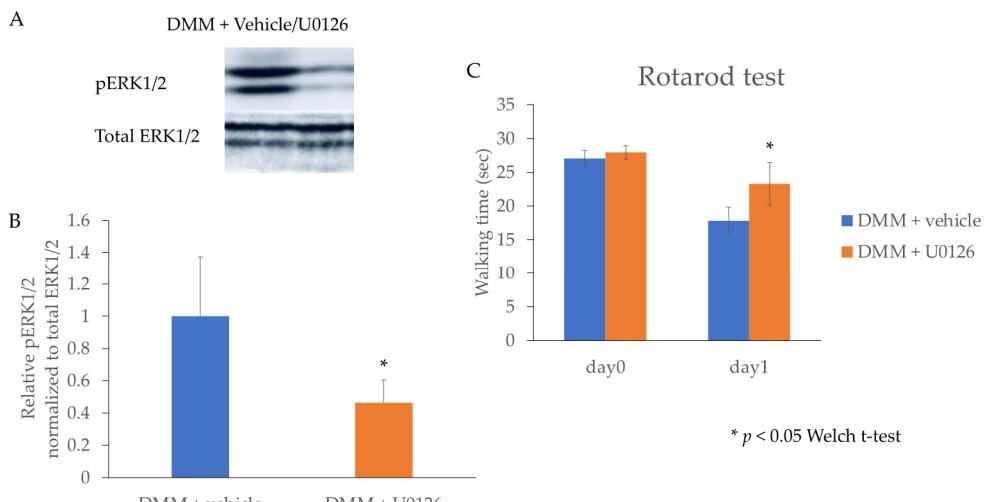

**Figure 6.** Expression of pERK1/2 and the rotarod performance test after the intrathecal injection of the MEK inhibitor U0126. (**A**) Immunoblot images of pERK1/2 and total ERK1/2 in the right dorsal horn of the spinal cord 1 day after surgery. (**B**) The quantified data of the immunoblot for pERK1/2 normalized to total ERK1/2. Each $n = 5$, * $p < 0.05$. (**C**) Walking time in the rotarod performance test. Each $n = 5$, * $p < 0.05$. All data are presented as mean ± standard deviation. pERK, phosphorylated extracellular signal-regulated kinase; DMM, destabilization of the medial meniscus.

## 4. Discussion

This study demonstrated that rats exhibited locomotive dysfunction in the acute phase after DMM surgery and in the chronic phase due to the development of KOA. Oral administration of BO improved this disability and suppressed the phosphorylation of ERK1/2 in the dorsal horn of the spinal cord.

Studies have shown that the DMM rat model exhibits osteoarthritic changes within 4 weeks after surgery [17,29]. Surgical intervention and KOA development induce knee pain and decreased latency to fall off the rotarod apparatus. The rotarod test is an established method to measure the ability of coordinated movement, and it can also be used in measuring pain-related locomotive disability [25]. Interestingly, in this present study, the rotarod test revealed that the latency to fall off the rotarod apparatus decreased not only in the chronic phase when KOA changes have occurred but also in the acute phase a few days after the DMM surgery. This result suggests that the postoperative acute pain caused by the DMM surgery hampered the locomotive function of rats. No difference was observed in the rotarod test between each group 7 days after DMM induction, which may be due to the healing of the surgical wound.

It has been reported that ERK1/2 phosphorylation in the dorsal horn of the spinal cord is involved in the development of pain in the postoperative pain model [21], and it has also been related to neuropathic pain and aggravated pain-related behavior [28,30]. Therefore, ERK1/2 phosphorylation in the dorsal horn of the spinal cord may lead to not only chronic pain, but also pain sensation and allodynia through various nociceptive stimuli. The stage in this present study is divided into acute phase 1 day after DMM induction and chronic phase 4 weeks after the surgery. In both phases, the phosphorylation of ERK1/2 in the

dorsal horn of the spinal cord was accelerated, indicating acute and chronic postoperative pain conditions. Intrathecal catheter injection of U0126, MEK inhibitor against ERK1/2 phosphorylation, was able to improve the rotarod test performance (Figure 6C). The phosphorylation of ERK1/2 induced by MEK may play a crucial role in pain-related behavior in the DMM model as well as in the postoperative pain model [21] and the chronic constriction injury model [28]. Therefore, this present study has demonstrated a novel therapeutic effect of BO, wherein the oral administration of BO reduced postsurgical or KOA-associated pain by inhibiting the phosphorylation of ERK1/2.

The rotarod test performed on day 1 showed that BO administration recovered the rotarod test results to $26.7 \pm 6.7$ s (Figure 3); however, the administration of MEK inhibitor recovered the results to only $23.3 \pm 2.0$ s (Figure 6). Therefore, BO may possess other analgesic mechanisms in addition to the inhibition of ERK1/2 phosphorylation. Fujitsuka et al. [15] reported that BO inhibited IL-1β secretion as well as joint fluid retention in the KOA rat model. Among the crude drugs present in BO, *Astragalus* root [31], *Glycyrrhiza* [32], and ginger [33] have been reported to exert analgesic effects against neuropathic pain. Kato et al. [34] demonstrated the inhibition of the calcium-activated chloride channel TMEM16A using liquiritigenin derived from *Glycyrrhiza*. Isoliquiritigenin, which is also derived from *Glycyrrhiza*, is known to inhibit voltage-gated sodium (Nav) channels in peripheral nociceptive neurons [35] and to exert an antagonistic effect on NMDA receptors in the spinal cord [36]. Furthermore, glycyrrhizin and its metabolite, 18β-glycyrrhetinic acid, and a compound found in *Glycyrrhizae* radix enhanced glutamic acid uptake through glutamate transporter 1 (GLUT-1) in astrocytes [37]. Considering these findings, the analgesic effect of BO is newly clarified through the inhibitory effect of ERK1/2 phosphorylation in the dorsal horn of the spinal cord, in addition to the anti-inflammatory and antinociceptive effect in the peripheral tissue. However, the detailed mechanism underlying the action of BO in the inhibition of MEK remains unclear. Thus, further investigation is required to elucidate the mechanism underlying the analgesic effect of BO against postsurgical or post-traumatic pain.

## 5. Conclusions

We explored the analgesic effect of BO on rats with DMM-induced KOA. BO exerted an analgesic effect in both the acute phase with post-traumatic injury and the chronic phase when KOA is progressing. We have clarified postoperative analgesia and inhibition of pain sensitization through the suppression of ERK1/2 phosphorylation in the dorsal horn of the spinal cord in this study.

**Author Contributions:** Conceptualization, Y.K. and T.O.; methodology, T.O., H.I. and N.A.; validation, T.O.; formal analysis: Y.K.; investigation, Y.K., T.O., M.T., T.K. and K.Y.; data curation, T.O. and H.I.; resources, H.I.; writing—original draft, Y.K.; writing—review and editing, T.O., H.I., N.A., M.T., T.K., K.Y., K.K. and M.S.; visualization, Y.K.; supervision, K.K. and M.S.; project administration, M.S.; funding acquisition, T.O. All authors have read and agreed to the published version of the manuscript.

**Funding:** This work was not supported by any of the funding associations.

**Institutional Review Board Statement:** All experimental procedures were approved by the Institutional Ethics Committee for Care and Use of Animals of Showa University (certificate number: 03053, date of approval: 1 April 2021).

**Data Availability Statement:** Data from this study will be available upon request for disclosure.

**Acknowledgments:** The authors are grateful to Tsumura & Co. for generously providing BO.

**Conflicts of Interest:** The authors declare no conflict of interest.

## Abbreviations

BO, boiogito; DAPI, 4′,6-diamidino-2-phenylindole; DMM, destabilization of the medial meniscus; DMSO, dimethyl sulfoxide; EDTA, ethylenediaminetetraacetic acid; ERK, extracellular

signal-regulated kinase; KOA, knee osteoarthritis; MEK, mitogen-activated protein kinase; MMTL, medial meniscotibial ligament; OARSI, Osteoarthritis Research Society International; PBS, phosphate buffer saline; PVDF, polyvinylidene difluoride; SDS, sodium dodecyl sulfate; SDS-PAGE, sodium dodecyl sulfate-polyacrylamide gel electrophoresis; TBST, Tris-buffered saline buffer and Tween 20.

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
