# Peer review of "Analgesic Effect of Boiogito, a Japanese Traditional Kampo Medicine, on Post-Traumatic Knee Osteoarthritis through Inhibition of ERK1/2 Phosphorylation in the Dorsal Horn of the Spinal Cord"

_applsci, doi:10.3390/app11188421_

Round 1

Reviewer 1 Report

The study is well planned and modern methodology has been used to implement this scientific idea. The results obtained are well described and  from a practical point of view much of interest. 

Author Response

Comment

  1. The study is well planned, and modern methodology has been used to implement this scientific idea. The results obtained are well described and from a practical point of view much of interest.

Reply

We are extremely grateful to you for your kind appreciation of our manuscript.

Reviewer 2 Report

Authors provided an interesting manuscript, especially for the application proposed. 

The scientific soundness is good. 

I suggest the addition of an abbreviation list to collect all the acronyms user in this work, according to this journal guidelines.

 The introduction is well written, but the definition of the aims of this paper should be improved.

A good presentazione of results and discussion.

I suggest minor revisions. 

Thank you

Author Response

Comments

  1. Authors provided an interesting manuscript, especially for the application proposed. The scientific soundness is good. I suggest the addition of an abbreviation list to collect all the acronyms user in this work, according to this journal guidelines.

Thank you for this suggestion. We apologize for the overlook with respect to the journal recommendation. As per your suggestion, we have added an abbreviation list in the revised manuscript.

  1. The introduction is well written, but the definition of the aims of this paper should be improved. A good presentation of results and discussion. I suggest minor revisions.

Thank you for your valuable comment. We are glad to receive your appreciation and hear that our manuscript needs only minor revisions. Thank you for pointing that the aim of this study should be written in more detail. We have revised it accordingly.

Reviewer 3 Report

Comments to the Authors of manuscript number: applsci-1352727 entitled “Analgesic Effect of Boiogito, a Japanese Traditional Kampo Medicine, on Posttraumatic Knee Osteoarthritis through Inhibition of ERK1/2 Phosphorylation in the Dorsal Horn of the Spinal Cord”.

Authors have presented a study performed on rats which related to the osteoarthritis-associated pain. The study is arranged well, it includes four groups which allow to compare obtained results in correct manner. Traditional medicine is very interesting. The number of studies performed to investigate Boiogito is not too big for the last decade. All these studies have showed very useful properties of this herbs. However, Authors should stronger underline what They want to present (against pain or KOA development).

  1. L 53 – improve inflammation?? It is not correct phrase. It should be corrected.
  2. L 59 – reference should be given
  3. L 71 – against knee injury or the osteoarthritis-associated pain? These are two different subjects.
  4. The part 2.1. -should be described in more details. There is no information how many rats were in the study.
  5. L 106 – what kind of extract was used? What is the main composition of this herbal? Is this composition dependent of season or geographical place? All this information should be clarified.
  6. L 273- There is different information than in abstract. The main goal is to decrease the pain not prevent before KOA development. It should be more clarified.
  7. Conclusion cannot include references. This part summarizes obtained results. It is not a discussion.

Author Response

Authors have presented a study performed on rats which related to the osteoarthritis-associated pain. The study is arranged well, it includes four groups which allow to compare obtained results in correct manner. Traditional medicine is very interesting. The number of studies performed to investigate Boiogito is not too big for the last decade. All these studies have showed very useful properties of this herbs. However, Authors should stronger underline what They want to present (against pain or KOA development).

  1. L 53 – improve inflammation?? It is not correct phrase. It should be corrected.

Thank you for pointing it up. We would like you to check the revised phrase.

  1. L 59 – reference should be given

Thank you for your valuable suggestions. The content described in L59 has been taken up from the reference cited in L60. Therefore, the sentences in L59–60 have been combined into one sentence and the reference for these contents has been shown as [17].

  1. L 71 – against knee injury or the osteoarthritis-associated pain? These are two different subjects.

Thank you for raising this point. In this study, we examined the analgesic effect of BO on both acute postoperative pain 1 day after DMM and KOA-associated pain that occurred when KOA developed 28 days after DMM. We agree that the phrase analgesic effect of BO against “knee injury” was certainly inappropriate. We have now re-written this phrase as “analgesic effect of BO on acute postoperative pain.” Please check.

  1. The part 2.1. -should be described in more details. There is no information how many rats were in the study.

In this study, a total of 50 rats, including 40 rats in Experiment 1 and 10 rats in Experiment 2, were used. We have added this information in the revised manuscript.

  1. L 106 – what kind of extract was used? What is the main composition of this herbal? Is this composition dependent of season or geographical place? All this information should be clarified.

In this study, we used boiogito (BO) manufactured by Tsumura & Co., Ltd. BO is marketed as a prescription drug. Most of the crude drugs contained in BO are cultivated in China, but because of the stable distribution route, they can be supplied throughout the year. BO contains a dry extract of a mixed drug substance consisting of 5.0 g of Sinomenium stem, 5.0 g of Astragalus root, 3.0 g of Atractylodes lancea rhizome, 3.0 g of Jujube, 1.5 g ofGlycyrrhiza, and 1.0 g of ginger. These herbs are mixed and extracted with purified water at 95.1°C for 1 h, and the soluble extract is then separated from the insoluble residue and dried by removing water under reduced pressure. We have added this information in the revised manuscript. All crude drugs are listed in the 17th edition of the Japanese Pharmacopoeia [22]. These varieties, sites, and the proportion of the main pharmacological active ingredients contained in each crude drug are strictly defined. Therefore, even if these producing districts and harvest seasons are different, there is no big difference in the effect. We have added that contents.

  1. Pharmaceutical and Medical Device Regulatory Science Society of Japan. Japanese pharmacopoeia, 17th ed, English version; Yakuji Nippo. Tokyo, Japan, 2016.

  1. L 273- There is different information than in abstract. The main goal is to decrease the pain but not to prevent before KOA development. It should be more clarified.

As indicated, I agree that it was an overrated expression. Through this study, we revealed that BO exerts an analgesic effect by suppressing the phosphorylation of ERK in the dorsal horn of the spinal cord. We have focused on this aspect and revised the text accordingly.

  1. Conclusion cannot include references. This part summarizes obtained results. It is not a discussion.

Thank you for raising this point. We have deleted the unnecessary sentences as suggested.

Round 2

Reviewer 3 Report

i have no additional comments

Author Response

We are extremely grateful to you for your kind appreciation of our manuscript.